# B-Spar: Bayesian Sparse-Reward Modeling for RL-based Image Editing

Shusong Xu [1]  Peiye Liu [2]  Yongbin Liu [1]  Bangjie Yin [1]  Tianyi Zheng [1]  Zhaomang Sun [1]  Zhenyu Chen [1]
Peng-Tao Jiang [1]  Jian Zhang [1]  Yuzhao Wang [1]  Zhen Gu [2]  Jinwei Chen [1]  Bo Li [1]

## Abstract

Autonomous image-editing agents powered by multimodal large language models (MLLMs) improve transparency and controllability by translating high-level instructions into tool-mediated edit sequences, but training such agents with reinforcement learning often relies on dense proxy rewards (e.g., incremental image-quality score gains) to compensate for sparse human feedback. When these proxies overvalue small local changes, the resulting optimization signal can be dominated by numerically measurable yet perceptually negligible edits, biasing policy gradients toward proxy artifacts rather than meaningful progress. We propose B-Spar, a reward-centric Reinforcement Learning framework for perceptually aligned image retouching under sparse feedback that combines prior-guided trajectory sampling to reduce inefficient exploration, Bayesian reward modeling to densify sparse binary feedback into a stable training signal, and anchor-regularized policy optimization to steer updates toward high-reward regions while preventing early mode collapse. Experiments on public benchmarks demonstrate that B-Spar improves perceptual quality and metric alignment with stable training and competitive inference efficiency over strong prompt-based and training-based baselines. Notably, it outperforms AIGC-based baselines by over 95% in perceptual quality, achieving an improvement of approximately 33.5% over the state-of-the-art.

## 1. Introduction

Autonomous agents powered by multimodal large language models (MLLMs) (Yuan et al., 2025; Durante et al., 2025; Yao et al., 2022) are rapidly gaining traction in image editing.

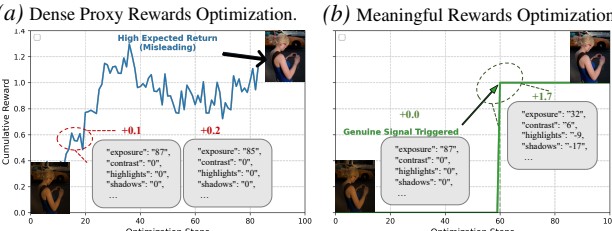

*Figure 1.* Illustration of proxy reward misalignment preventing real perceptual gains. (a) Dense proxy rewards (blue) yield misleadingly high cumulative returns via frequent, trivial parameter changes (*e.g.*, +0.1, +0.2) without perceptual gain. (b) Meaningful rewards (green) remain sparse, triggering a sharp jump (*e.g.*, +1.7) upon genuine visual improvement.

By leveraging a "Perceive-Reason-Act-Reflect" cognitive workflow, these frameworks logically decompose high-level editing directives into executable, fine-grained tool chains, thereby eschewing direct pixel generation. Crucially, unlike the end-to-end generative paradigms inherent in models such as Gemini (Comanici et al., 2025) and Qwen (Yang et al., 2025b), this agent-centric methodology introduces transparency to the black-box editing process while substantially improving control precision and fidelity.

Within this landscape, methodologies for image retouching agents can be broadly categorized into prompt-based and training-based approaches. Prompt-based agents (Chen et al., 2025; Ye-Bin et al., 2025; Liang et al., 2025) exploit sophisticated Chain-of-Thought (CoT) prompting to inject domain-specific heuristics into general-purpose MLLMs. Conversely, training-based methods (Dutt et al., 2025; Chang et al., 2025; Lin et al., 2025) typically employ Supervised Fine-tuning (SFT) or Reinforcement Learning (RL) to optimize task-specific performance.

Despite their promise, current RL-based editing agents are often trained with dense rewards proxying sparse human feedback such as small gains in image-quality scores, because these signals are frequent and facilitate stable policy optimization as illustrated in Figure 1. However, when the reward function gives high scores to small local changes, it effectively creates a reward distribution that puts high probability on behaviors yielding numerically measurable yet perceptually negligible changes. As a result, the learn-

[1]vivo Mobile Communication Co., Ltd, Shanghai, China [2]Independent Researcher, Beijing, China. Correspondence to: Bo Li <libra@vivo.com>.

*Proceedings of the 43rd International Conference on Machine Learning*, Seoul, South Korea. PMLR 306, 2026. Copyright 2026 by the author(s).

ing signal becomes dominated by low-value behaviors, and policy gradients are biased toward optimizing proxy artifacts instead of genuine progress. This motivates reward modeling that focuses on optimization signal on trajectories corresponding to meaningful improvements, while reducing fake rewards induced by trivial proxy noise. To address this, we need a reward system that focuses on meaningful improvements while minimizing the impact of irrelevant fluctuations in behavior.

Given this, we propose **B-Spar** (Bayesian Sparse-reward Modeling), a reward-centric RL system designed for perceptually-aligned image editing agents. B-Spar mitigates the challenge through three key mechanisms. (1) The Prior-Guided Trajectory Sampling uses domain knowledge to initialize trajectories, avoiding the wasted effort of blind search when hunting for rare positive states. (2) The Bayesian Reward Modeling augments and densifies sparse binary rewards using Bayesian optimization to keep training stable with limited data. (3) The Anchor-Regularized Policy Optimization introduces hint-conditional anchors to guide the policy toward high-reward regions while constraining updates to prevent mode collapse during early training. Our main contributions are summarized as follows:

- We propose a novel reward-centric RL framework to rectify the optimization bias where policy gradients focus on reward-proxy artifacts instead of real progress, especially in high-dimensional image editing tasks.

- We introduce a Bayesian Reward Densification mechanism that transforms sparse binary feedback into a dense, reliable optimization signal.

- We conduct extensive experiments on public benchmarks, showing that our method achieves state-of-the-art performance, surpassing baselines in perceptual quality, metric alignment, and inference efficiency.

## 2. Related Work

**Agentic MLLM.** Building on progress in reasoning-enhanced MLLMs (Zhuang et al., 2025; Wang et al., 2025; Huang et al., 2025) and tool-augmented interaction, a growing body of work has shifted from workflow-bound systems toward agentic MLLMs, which frame MLLMs as autonomous decision-makers capable of planning, reflection, memory, tool use, and environment interaction (Team et al., 2025b;c; Geng et al., 2025). Compared to pipeline-based agents, agentic MLLMs adaptively generate and revise action plans based on historical context and environmental feedback, enabling more flexible and general-purpose behavior across tasks and domains. Online reinforcement learning plays a central role in enabling such adaptive agentic behavior and aligning (multimodal) language models with human preferences (Brandizzi, 2024), as well as in visual generation systems (Team et al., 2025a). Existing approaches can be broadly categorized by their supervision signals into three paradigms. Learning from human feedback (RLHF) relies on learned reward models trained from human annotations (Ouyang et al., 2022; Rafailov et al., 2023; Touvron et al., 2023), but incurs high annotation costs and is vulnerable to reward hacking and limited generalization under distribution shift. Reinforcement learning with verifiable rewards (RLVR) replaces learned rewards with explicit verification signals, enabling stable optimization in tasks with well-defined correctness criteria (Sang et al., 2025; Roussille, 2024; Li, 2022; Ibrahim et al., 2024; Jia et al., 2025; Groß, 2024), yet remains restricted to closed-form or rule-based domains. Reinforcement learning from internal feedback (RLIF) leverages model-generated evaluations or critiques to enable scalable self-improvement (?Khanal et al., 2026; Kumar et al., 2025; Kourani et al., 2025; Song et al., 2024), but often suffers from self-deception and overconfidence, leading to unstable training dynamics in online settings. While RLVR (Jin et al., 2025) provides stable signals in closed domains and RLIF (Yuan et al., 2024) enables autonomous scaling, they both lack the ability to actively seek external information or perform extensive search over the solution space. To address this, Online Search-Augmented RL (Jin et al., 2025) introduces test-time computation and real-time retrieval into the RL loop, bridging the gap between internal reasoning and external verification.

**Instruction-Based Image Editing.** Instruction-driven image editing has become a cornerstone of modern artificial intelligence, enabling models to modify visual content in a controllable, semantically grounded, and aesthetically meaningful manner. Recent progress has led to unified image editing models that jointly integrate multimodal understanding and content generation, achieving strong performance in both instruction comprehension and creative synthesis. Representative closed-source systems include GPT-4o (OpenAI, 2024) and Nano-Banana (Google, 2025), while notable open-source efforts such as Bagel (Li et al., 2025a), step1x-edit (Liu et al., 2025), and Qwen Omni (Xu et al., 2025) further demonstrate the scalability and generality of this paradigm. Beyond unified architectures, a growing line of work explores agentic, tool-integrated editing systems that emulate human designer workflows by decomposing complex instructions, selecting appropriate tools, and coordinating multi-step edits for fine-grained aesthetic refinement. Exemplary approaches include JarvisArt (Lin et al., 2025), PhotoArtAgent (Chen et al., 2025), and MonetGPT (Dutt et al., 2025). Despite these advances, two fundamental challenges remain unresolved: (1) susceptibility to hallucinations under complex or ambiguous instructions, even when explicit textual reasoning is incorporated, and (2) reliance on fixed reward models in reinforcement learning, which are

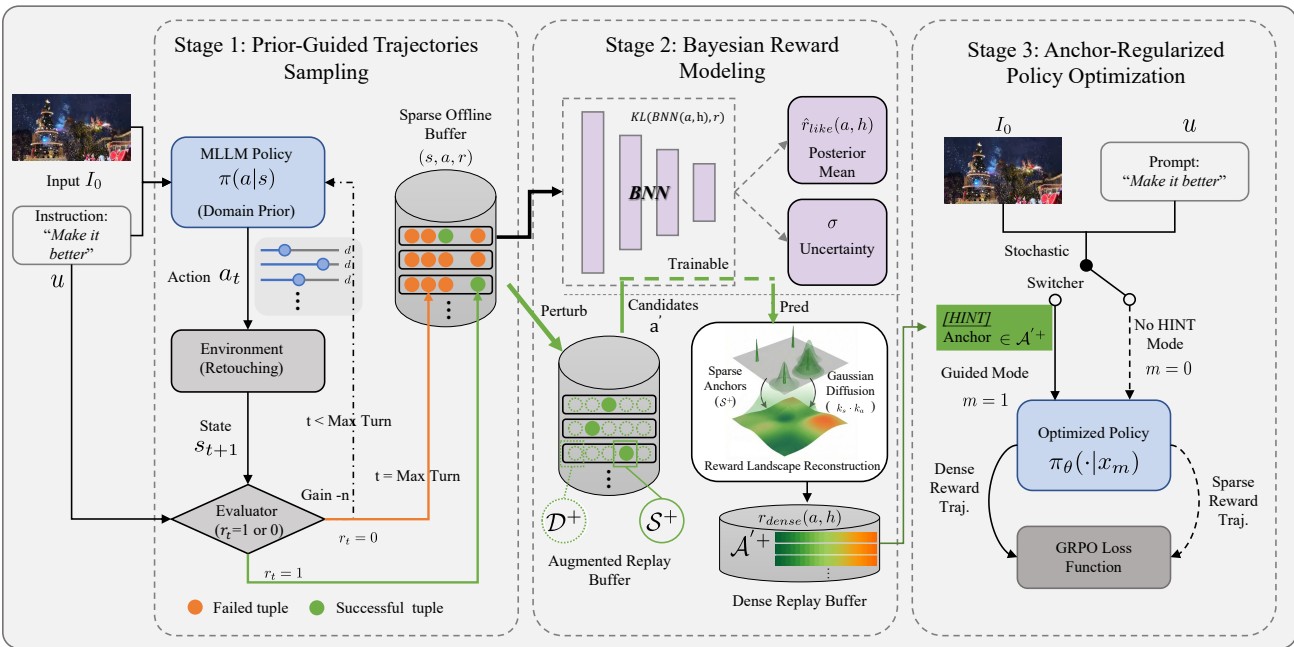

*Figure 2.* Overview of our framework. Our approach comprises three main stages: (1) Prior-Guided Trajectory Sampling uses domain knowledge to initialize trajectories, avoiding the wasted effort of blind search when hunting for rare positive states. (2) Bayesian Reward Modeling augments and densifies sparse binary rewards using Bayesian Network. (3) The Anchor-Regularized Policy Optimization introduces hint-conditional anchors to guide the policy toward high-reward regions while constraining updates to prevent mode collapse during early training.

prone to reward hacking and often fail to generalize beyond their training distributions.

**Sparse Rewards Optimization.** Sparse rewards frequently induce instability or stagnation in the learning process, primarily due to the inefficiency of random exploration within vast state spaces. Some approaches have addressed this challenge by introducing auxiliary rewards (Li et al., 2025b) or employing reward shaping (Hu et al., 2020). Furthermore, intrinsic motivation mechanisms (Lai et al., 2023) have been proposed to encourage broader state coverage, often utilizing exploration incentives based on prediction error or information gain (Bigazzi et al., 2022; Devidze et al., 2022). Structurally, hierarchical reinforcement learning (Ji et al., 2025) mitigates reward sparsity for high-level policies by decomposing long-horizon goals into learnable sub-goals. Similarly, goal-conditioned learning and hindsight replay techniques transform unrewarded interactions into effective supervision via trajectory relabeling (Luo et al., 2024), demonstrating significant robustness in sparse reward settings. In the context of language and multimodal agents, recent methodologies leverage human feedback, preference modeling (Rafailov et al., 2023), and online policy optimization (Wu et al., 2024) to provide structured learning signals, significantly bolstering training stability while circumventing the dependency on the dense, manually engineered rewards often found in prior works.

## 3. Methods

### 3.1. Preliminary

**Problem Formulation.** We formulate instructional image editing as a sequential decision process, where the MLLM agent iteratively refines the image $I_t$ based on the user instruction $u$. At each step $t$, the agent observes the current state $s_t = (I_t, u)$ and predicts an action $a_t = (\mathrm{m}_t, \mathrm{p}_t)$, which deterministically updates the image through the retouch engine: $I_{t+1} = \mathcal{E}(I_t, a_t)$. To enforce perceptible validity, we replace continuous feedback with a binary reward that filters out negligible improvements. We define the task reward $r_{\text{task}}(s, a)$ as:

$$r_{\text{task}}(s, a) = \begin{cases} 1, & \text{if } \mathcal{Q}(\mathcal{E}(I, a), u) > \mathcal{Q}(I, u) + \delta, \\ 0, & \text{otherwise.} \end{cases}$$

where $\mathcal{Q}$ is the quality metric and $\delta$ represents the minimum improvement margin. This discrete formulation ensures that the optimization objective $J(\theta) = \mathbb{E}_{\pi_\theta}[r_{\text{task}}]$ is driven solely by semantically meaningful edits.

**The Challenge of Reward Sparsity.** The binary reward introduces a severe optimization bottleneck: *extreme sparsity* within the high-dimensional action space. The set of valid actions is defined as: $\mathcal{A}^+ = a \in \mathcal{A} \mid \mathcal{Q}(\mathcal{E}(I, a), u) > \mathcal{Q}(I, u) + \delta$. Navigating to $\mathcal{A}^+$ presents a fundamental dilemma. The action space involves

a joint distribution of discrete tool selection and continuous parameter adjustment. For an edit to be valid, the agent must not only select the semantically correct tool but also pinpoint specific parameter values that satisfy the strict margin $\delta$. Consequently, valid edits become rare events. Geometrically, for a given state $s$, the valid subset $\mathcal{A}_s^+ = \{a \mid r_{\text{task}}(s, a) = 1\}$ forms a negligible volume within the solution space. As perceptual improvement requires precise coordination, if the valid range per dimension is a fraction $\epsilon$ of the total range $L$, the sampling probability scales as $(\epsilon/L)^d$. This exponential decay with dimensionality implies that naive exploration policies almost never sample from $\mathcal{A}^+$, resulting in vanishing gradients and policy collapse before valid regions are discovered.

### 3.2. Method Overview

In this section, we introduce B-Spar, a reward-centric framework for training image editing agents under sparse human feedback. B-Spar comprises three components: Prior-Guided Trajectories Sampling, Bayesian Reward modeling, and Anchor-Regularized Policy Optimization, which jointly provide efficient exploration and stable optimization while emphasizing trajectories corresponding to meaningful improvements.

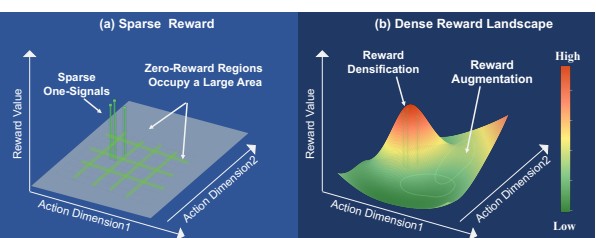

*Figure 3.* Visualization of the Reward Landscape Densification.

**Prior-Guided Trajectories Sampling.** We formulate image editing as a short-horizon Markov Decision Process (MDP) with a maximum horizon $T = 2$. At each timestep, the MLLM acts as a stochastic policy $\pi(a_t \mid s_t)$, where the state is defined as $s_t = (I_t, u)$ with the current image $I_t$ and the high-level instruction $u$. Unlike random exploration in high-dimensional action spaces, we leverage the MLLM's domain prior to bias exploration toward candidate trajectories that are more likely to yield high rewards. Because valid edits satisfying strict binary criteria are rare, directly optimizing the policy at this stage can incur high-variance learning signals. We therefore focus on *exploration* only, collecting a trajectory buffer without updating parameters. At each step, the policy samples an action $a_t$ (an operation with its parameters) and transitions to $s_{t+1}$. We then evaluate the resulting edit using strict binary criteria (e.g., human feedback or thresholded metrics). If the edit is deemed a valid improvement, we assign a terminal reward $r = 1$; otherwise $r = 0$. With $T = 2$, the agent is allowed at most

one additional attempt before termination. This process yields a dataset of logged tuples $(s_t, a_t, r)$, where $r \in {0, 1}$ is the terminal reward assigned at the end of the episode.

---

**Algorithm 1** Bayesian Sparse-Reward Modeling

---

**Require:** Images $\{I\}$, instructions $\{u\}$, MLLM policy $\pi_\phi$, reward $r(\cdot)$, max Turn $T$, dropout $p_{\text{drop}}$
**Ensure:** Domain-optimized policy $\pi_\theta$
1: Initialize trajectory dataset $\mathcal{D}_0 \leftarrow \emptyset$
2: **for** each $(I, u)$ **do**
3:     $s_0 \leftarrow (I, u)$
4:     **for** $t = 0$ to $T - 1$ **do**
5:         Sample action $a_t \sim \pi_\phi(a_t \mid s_t)$
6:         Apply action $I' \leftarrow \mathcal{E}(I, a_t)$
7:         Compute reward $r_t \leftarrow r(I', u)$
8:         $\mathcal{D}_0 \leftarrow \mathcal{D}_0 \cup (s_t, a_t, r_t)$
9:         **if** $r_t = 1$ **then**
            break
10:         **end if**
11:         $s_{t+1} \leftarrow (I', u)$
12:     **end for**
13: **end for**
14: Perform Bayesian Reward Modeling to obtain $\mathcal{D}_{\text{aug}}$
15: Densify rewards using Gaussian kernels:

$$r_{\text{dense}}(a, h) \leftarrow \frac{\sum_{(a', h') \in \mathcal{D}^+} \hat{r}_{\text{post}}(a', h') \, k_h(h, h') \, k_a(a, a')}{\sum_{(a', h') \in \mathcal{D}^+} k_h(h, h') \, k_a(a, a') + \epsilon}$$

16: Initialize policy dataset $\mathcal{D}_\pi \leftarrow \mathcal{D}_{\text{aug}}$
17: **for** each training step **do**
18:     Sample mask $m \sim \text{Bernoulli}(1 - p_{\text{drop}})$
19:     Set input context

$$x_m = \begin{cases} (u, I, [\texttt{HINT}], a^+) & m = 1 \\ (u, I) & m = 0 \end{cases}$$

20:     Sample action $a \sim \pi_\theta(\cdot \mid x_m)$
21:     Update $\theta \leftarrow \arg\max_\theta \mathbb{E}[R_{\text{dense}}(a)]$
22: **end for**
    Return $\pi_\theta$

---

**Bayesian Reward Modeling.** After collecting an interaction dataset $\{(a_i, I_i, r_i)\}_{i=1}^N$ with binary success labels $r_i \in \{0, 1\}$, we construct a dense training signal from sparse feedback to support stable policy updates. We first extract a compact image descriptor $h = \phi(I)$ using RGB intensity histograms, which encode coarse statistics such as brightness and color. We then train a Bayesian neural network (BNN) to model the success likelihood $p_\theta(r = 1 \mid a, h)$. Monte Carlo sampling of BNN weights yields an empirical predictive probability $\hat{r}_{\text{like}}(a, h) = \frac{1}{K} \sum_{k=1}^K p_{\theta_k}(r = 1 \mid a, h)$, which we treat as a soft success observation. To regularize estimates under limited data, we impose a Beta prior $\text{Beta}(\alpha_0, \beta_0)$ on the Bernoulli suc-

cess probability and compute the conjugate posterior mean $\hat{r}_{\text{post}}(a, h) = \frac{\alpha_0 + \hat{r}_{\text{like}}(a,h)}{\alpha_0 + \beta_0 + 1}$.

Starting from observed successful samples $(a, h) \in \mathcal{S}^+$, we generate candidate actions $a'$ via Gaussian perturbations for local exploration in the action space, and evaluate each candidate $(a', h')$ using $\hat{r}_{\text{post}}(a', h')$ (with $h' = \phi(I')$). We retain candidates whose posterior success probability exceeds a threshold and assign them soft pseudo-labels given by their posterior means, forming an augmented set $\mathcal{D}^+$.

Finally, we densify supervision by diffusing these pseudo-labels to nearby points with Gaussian kernels:

$$r_{\text{dense}}(a, h) = \frac{\sum_{(a',h')\in\mathcal{D}^+} \hat{r}_{\text{post}}(a', h')\, k_h(h, h')\, k_a(a, a')}{\sum_{(a',h')\in\mathcal{D}^+} k_h(h, h')\, k_a(a, a') + \epsilon}. \tag{1}$$

Here $k_h(h, h') = \exp(-\|h - h'\|^2/(2\sigma_h^2))$, $k_a(a, a') = \exp(-d(a, a')^2/(2\sigma_a^2))$, and $d(\cdot, \cdot)$ is a distance metric in the action space. This Bayesian model provides uncertainty-regularized success estimates, while kernel diffusion yields smooth dense rewards for learning under sparse feedback.

**Anchor-Regularized Policy Optimization.** Following reward densification, we use the collected actions to guide policy learning in the MLLM. We introduce a Anchor-Regularized Policy Optimization (AQPO) strategy, where an anchor action $a^+$ is drawn from $\mathcal{D}^+ \cup \mathcal{S}^+$ and provided as a hint to the policy. However, naively conditioning the policy on anchors creates a risk of over-reliance, where the agent learns to exploit the anchor as a trivial shortcut rather than reasoning through the editing instructions. To counteract this tendency, we implement a *signal dropout* mechanism.

Let $m \sim \text{Bernoulli}(1 - p_{\text{drop}})$ be a binary mask and define the input context $x_m$ as

$$x_m = \begin{cases} (u, I, [\text{HINT}], a^+) & \text{if } m = 1, \\ (u, I) & \text{if } m = 0, \end{cases} \tag{2}$$

where $[\text{HINT}]$ is a special delimiter token. When $m = 1$, the model performs guided refinement; when $m = 0$, it acts autonomously. We then optimize the policy by maximizing the expected densified reward:

$$\theta^* = \arg\max_\theta \, \mathbb{E}_{a\sim\pi_\theta(\cdot|x_m)}\big[r_{\text{dense}}(a, h)\big], \tag{3}$$

where $r_{\text{dense}}(a, h)$ is defined in the previous section.

# 4. Experiments

## 4.1. Experimental Setup.

**Training Strategy.** Our training process is divided into two stages. In the first stage, we conduct domain knowledge training to enable the MLLM to master image editing tools and understand the effects of tool parameters. The dataset,

experimental setup, and training procedures for this stage are detailed in Supplementary Material A.1. In the second stage, we employ the GRPO-based online reinforcement learning paradigm. During the sampling phase, we generate 32 candidates for each image; these candidates are then augmented and densified. Finally, we perform online policy optimization to refine the model's ability to generate effective image editing plans.

**Implementation Details**. We use Qwen3vl-2B as the base model (Yang et al., 2025a), fine-tuning it via LoRA within the Swift framework. The LoRA rank is 256 and the alpha is 512. The model is fine-tuned for 4 epochs with 10,000 training samples on three NVIDIA H20 GPUs (98 GB memory each). For GRPO-based online training, we utilize 12,000 samples with a total training time of approximately 42 hours. We deploy vLLM in server mode for efficient rollout data collection, performing importance sampling at the sequence level. The KL divergence coefficient is set to 0.01. During generation, we employ a temperature of 1.02, a top-p of 0.95, and a top-k of 20.

**Baselines.** We evaluate our image retouching agent against representative methods from three categories: *AIGC-based* image editing models, *training-based* agents, and *prompt-based* agents. The *AIGC-based* methods include *MagicBrush* (Zhang et al., 2023), *OmniGen* (Yang et al., 2025b), *Step1XEdit* (Liu et al., 2025), and *Instruct-P2P* (Brooks et al., 2023), which perform image editing in an end-to-end manner by directly generating edited images conditioned on textual prompts, without explicitly modeling iterative reasoning or tool-based decision processes. *JarvisArt* (Lin et al., 2025) is a representative *training-based* image editing agent. It frames image editing as a sequential decision-making process and learns an editing policy through task-specific training. *PhotoArt* (Chen et al., 2025)serves as a *prompt-based* agent. It performs iterative editing by leveraging large language models for planning and tool invocation, without the need for task-specific training. In addition, we employed Qwen3vl-30B (Yang et al., 2025a) using a prompt-based approach (labeled as *Qwen3vl-30B\** in the table) to call our tools, serving as a baseline.

**Evaluation Metrics.** We evaluate our method using both subjective and objective metrics on the public MIT-FiveK test set with 800 samples (Bychkovsky et al., 2011). The objective metrics are categorized into three groups. The first group consists of standard reference-based metrics (PSNR, SSIM, and LPIPS) computed using images retouched by "Expert B" as the ground truth, where higher PSNR and SSIM scores indicate better fidelity, while lower LPIPS values correspond to better perceptual similarity. The second group incorporates state-of-the-art no-reference metrics, specifically Artimuse (Cao et al., 2025), PQ (Ku et al., 2024), and HyperIQA (Su et al., 2020), to rigorously eval-

*Table 1.* Quantitative Comparison. **Red bold** indicates the best result, and **Green bold** indicates the second best.

| | Full Reference | | | No Reference | | | Efficiency | |
|---|---|---|---|---|---|---|---|---|
| | $SSIM_{\times 10^2}$ ↑ | PSNR ↑ | $LPIPS_{\times 10^2}$ ↓ | Artimuse ↑ | PQ ↑ | HyperIQA ↑ | Latency(s) ↓ | Param ↓ |
| | **AIGC-based** | | | | | | | |
| MagicBrush | 58.68 | 16.81 | 20.91 | 52.92 | 8.43 | 64.91 | **5.59** | **0.9B** |
| OmniGen | 64.21 | 17.26 | 41.85 | 56.20 | 8.15 | 67.53 | 87.20 | 3.8B |
| Step1XEdit | 67.05 | 17.33 | 43.69 | 58.23 | **8.73** | **68.63** | 158.65 | 12B |
| Instruct-p2p | 65.97 | 18.39 | 21.19 | 55.35 | 8.21 | 63.06 | 24.96 | **1.0B** |
| | **Prompt-based** | | | | | | | |
| PhotoArt | **68.41** | **20.75** | **1.55** | 58.25 | 8.70 | 65.36 | 35.92 | 175B+ |
| Qwen3vl* | 66.37 | **19.35** | 1.90 | 59.28 | 8.59 | 66.16 | 14.86 | 30.0B |
| | **Training-based** | | | | | | | |
| Jarvisart | 58.95 | 18.78 | 7.80 | **60.31** | 8.39 | 67.43 | 77.78 | 7.61B |
| Ours | **67.88** | 16.34 | **1.03** | **61.37** | **8.84** | **69.02** | **1.86** | 2.01B |

uate aesthetic quality in the absence of paired references. The third group comprises the end-to-end latency and the model parameter size. Additionally, we also evaluate the Artimuse metric on the MMArt-PPR10k dataset (Lin et al., 2025).

## 4.2. Comparison.

**Objective Comparison.** Table 1 presents a comprehensive quantitative comparison between our proposed method and state-of-the-art image editing approaches. Our method achieves scores of 1.03 and 67.88 on LPIPS and SSIM, respectively—metrics measuring fidelity to the original image. Specifically, we attain 1.03 on LPIPS (lower is better), which strongly correlates with human perception, representing a relative improvement of approximately 33.5% over the second-best method (PhotoArt, 1.55 ) and over 95% compared to AIGC baselines such as MagicBrush. For structural similarity (SSIM, higher is better), we achieve 67.88 , surpassing AIGC methods including Step1XEdit (67.05 ) and Qwen3VL-30B (66.37 ) by at least 1.2% . Although we obtain lower scores on PSNR metrics emphasizing pixel-level alignment (e.g., representation differences and expert ratings), the significant LPIPS improvement indicates that our model prioritizes perceptual realism and semantic fidelity over strict pixel-wise correspondence—a trade-off that better aligns with human visual preferences. Regarding no-reference quality assessment, which typically exhibits stronger correlation with subjective human judgment, our method consistently achieves state-of-the-art results. Specifically, we obtain the highest scores on ArtiMUSE (61.37 ), PQ (8.84 ), and HyperIQA (69.02 ), leading the second-best methods by 0.6% to 1.8%. This demonstrates the superior aesthetic quality and authenticity of our generated images. This performance is further validated on the MMArt-PPR10k dataset. As shown in Table 4, our method achieves

61.22, consistently outperforming both specialized editing agents and general-purpose AIGC diffusion-based baselines. The consistent margin of improvement aligns with our main results, demonstrating strong cross-dataset generalization. Beyond objective quality, our model achieves an exceptional balance between performance and efficiency. With a latency of only 1.86 seconds (requiring at most 2 iterations), our method is approximately $41.81\times$ faster than agent-based approaches (JarvisArt, 77.78 s) and achieves nearly two orders of magnitude lower latency than Step1XEdit (158.65s). Furthermore, with only 2.01 B parameters, our model is significantly lighter than large-scale baselines (e.g., 12 B, 30 B, and 175 B parameters), making it highly suitable for on-device real-time deployment without compromising perceptual performance. Our approach consistently outperforms existing methods on key perceptual metrics while achieving the lowest latency.

**Subjective Comparison.** Figure 4 illustrates the qualitative comparison, highlighting the alignment between our proposed method and human feedback. By assigning higher rewards to actions that yield substantial visual improvements, our approach incentivizes the exploration of action spaces that maximize subjective perceptual differences. Consequently, the model generates images that exhibit superior alignment with human aesthetic preferences. Specifically, our optimization process significantly enhances visual aesthetics and accentuates the primary subject. This strategy effectively bridges the gap between rigid numerical objectives and genuine human visual perception. To further validate these perceptual gains, we conducted a blind user study involving 10 participants. Evaluators were tasked with rating the overall visual aesthetic quality of all samples—including the original images—on a scale of 1 to 10, where 1 denotes the lowest quality and 10 the highest. As depicted in the top panel of Figure 4, our method achieved the highest mean

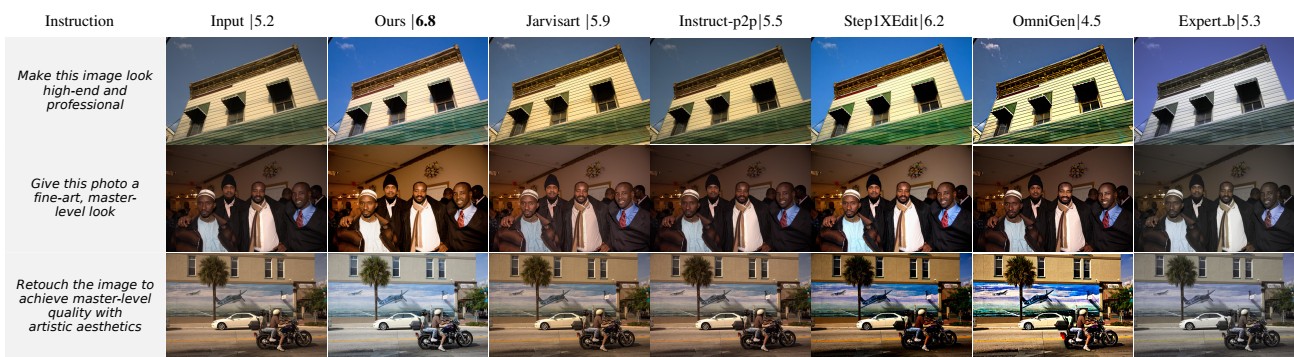

*Figure 4.* Visual comparison of different methods on MIT-fiveK. The scores annotated in the top row represent mean ratings from a user study ($N = 10$) conducted on a scale of 1–10 (where 1 denotes poor quality and 10 indicates high aesthetic quality). Our method achieves the highest average score of 6.8. By introducing a refined reward mechanism, our approach incentivizes the exploration of action trajectories that maximize subjective perceptual gains.

*Table 2.* Ablation of the Bayesian Reward Modeling. The table presents quality scores for the baseline, stepwise component integration, and the full training framework.

| Methods | HyperIQA $\uparrow$ | LPIPS$_{\times 10^2}$ $\downarrow$ |
|---|---|---|
| *Baseline* | | |
| Domain-adapted Qwen3vl-2B | 64.83 | 1.87 |
| *Only RL* | | |
| + Bayesian Augmentation | 68.58 | 1.22 |
| + Augmentation & Densification | 68.63 | 1.17 |
| *SFT + RL* | | |
| + Augmentation & Densification | 69.02 | 1.03 |

score of 6.8, outperforming images manually tuned by experts. Notably, the AIGC baseline, Step1XEdit, secured the second-highest user rating, despite exhibiting marginal disadvantages in full reference metrics. These results demonstrate that our method not only yields superior quantitative performance but also ensures the generated images align more closely with human preferences compared to standard agent-guided baselines.

### 4.3. Ablation Study

**Ablation of the Bayesian Reward Modeling.** We conducted an ablation study, with quantitative results summarized in Table 2. Using the Qwen3-VL-2B model as the baseline, the training dataset comprises approximately 500,000 tool usage and aesthetic understanding samples. At this stage, the model learns to use the tools but has not yet grasped how to enhance images based on image features, resulting in performance comparable to random behavior. Next, we collected around 5,000 positive feedback samples ($r = 1$) along with their corresponding actions by sampling from the AVA dataset and the MIT-FiveK training set. We then applied Bayesian optimization to increase the number of $r = 1$ samples to approximately 12,000. For reinforce-

ment learning (RL) training, we used GRPO with a binary reward setting. Additionally, we applied Gaussian densification within the Bayesian system to smooth the rewards of 0 and 1 for the GRPO-based RL training. Finally, we compared the model's performance after supervised fine-tuning (SFT) followed by GRPO-based RL training. The results, as shown in the table, demonstrate that guiding the model into high-value regions leads to the most significant performance improvement.

**Ablation with other RL methods.** To isolate the contribution of each design choice, we conduct ablation studies on the MIT-FiveK dataset (Table 3). We benchmark our method against two representative paradigms: (1) **Direct Optimization via Data Augmentation** (e.g., DPO), which achieves moderate perceptual gains (HyperIQA 66.56) but often compromises structural integrity in subtle; and (2) **Dense Reward Shaping** via trained reward models (e.g., employing GRPO with learned reward functions to densify binary 0-1 signals), which surprisingly achieves inferior image quality (HyperIQA merely 64.8)—even lower than the direct optimization baseline. Our high-value anchored reinforcement learning substantially outperforms both approaches, improving over the dense reward shaping baseline by approximately $6.5\%$ (from 64.8 to 69.02) and over direct optimization by $3.7\%$. By unifying Bayesian reward modeling with prior-guided sampling, our method resolves the dilemma between perceptual quality and structural fidelity, simultaneously achieving superior aesthetic scores (HyperIQA **69.02**) and pixel-level alignment (LPIPS **1.03**).

*Table 3.* Comparison with other RL method

| Methods | HyperIQA $\uparrow$ |
|---|---|
| Direct Optimization via Data Augmentation | 66.56 |
| GRPO based on the Reward Model | 64.80 |
| Our Unifying Bayesian reward modeling | 69.02 |

**Ablation of the Hyperparameters.** We evaluated the robustness of the B-Spar framework under different configurations of its core hyperparameters, including the minimum perceptible improvement threshold $\delta$ and the Gaussian kernel widths $\sigma_h$ and $\sigma_a$. Detailed values are reported in Appendix A.4.

First, we analyze the effect of the $\delta$ value used to convert a continuous function into a binary one. The threshold $\delta$ serves to filter out minor local fluctuations and high-frequency noise in the reward signal. We vary $\delta \in 1, 2, 3$ and observe the following HyperIQA performance: $68.75 \pm 4.48$ for $\delta = 1$, $69.02 \pm 4.07$ for $\delta = 2$, and $68.69 \pm 4.61$ for $\delta = 3$. All variations fall within the standard deviation range, indicating that, as long as trivial improvements are effectively filtered out, the framework exhibits strong stability across a moderate range of threshold values.

The Gaussian kernel widths determine the extent of the local reward diffusion mechanism. The analysis in Appendix A.4 presents a sensitivity analysis under different combinations of kernel widths. Within a moderate range $(\sigma_a, \sigma_h \in 0.5, 1.0)$, configurations consistently maintain robust performance with minimal variance ($68.85 \pm 0.16$). When the kernel size becomes too large (e.g., $2.0$), we observe a slight drop in average performance accompanied by increased variance. This aligns with theoretical expectations, as excessive smoothing of the reward landscape diminishes the ability to distinguish fine-grained editing operations.

*Table 4.* Evaluation on the MMArt-PPR10k dataset.

| Methods | Artimuse Score ↑ | Latency(s) ↓ |
|---|---|---|
| AIGC-based | | |
| MagicBrush | 47.92 | 5.59 |
| OmniGen | 53.65 | 87.20 |
| Step1XEdit | 55.64 | 158.65 |
| Instruct-p2p | 56.54 | 24.96 |
| Training-based | | |
| Jarvisart | 59.13 | 77.78 |
| Ours (B-Spar) | **61.22** | **1.86** |

### 4.4. The Analysis of the Data Distribution.

**Bayesian Optimization for Reward Distribution.** To address the challenge of sparse rewards in high-dimensional image editing, we aim to efficiently identify high-quality editing trajectories without exhaustive exploration. We employ a knowledge-enhanced MLLM to generate candidate actions offline. Rewards are computed based on relative improvement in an Image Quality Assessment (IQA) metric: we assign a reward of 1 if the IQA score(Su et al., 2020) of the edited image exceeds that of the original by a threshold $\delta$ (we set $\delta = 2$ to balance the scarcity of positive

signals against perceptual significance), and 0 otherwise. to evaluate the quality of generated actions, we measure the reward proportion (the fraction of high-reward samples) and the Diversity (to ensure broad exploration of the action space). the Diversity is quantified by flattening each function's parameters into a global vector (with missing functions zero-padded) and computing the average pairwise Euclidean distance between normalized vectors; higher values indicate broader coverage of the editing space. We compare three strategies on 5000 images: random sampling, vanilla offline rollout (MLLM-guided), and our Bayesian-augmented rollout. As shown in Table 5, random sampling achieves the highest diversity (2.983) but nearly zero reward proportion (0.004), confirming that blind exploration rarely discovers effective editing operations. Vanilla offline rollout improves reward proportion to $9.3\%$ while maintaining moderate diversity (1.608), yet still wastes over $90\%$ of samples on low-value actions. Our Bayesian augmentation significantly enhances sample efficiency, boosting the reward proportion to $\mathbf{23.6}\%$—a $153.8\%$ **relative gain** over vanilla rollout and $59\times$ **improvement** over random sampling. Notably, diversity also increases by $6.5\%$ (1.712 vs. 1.608), demonstrating that Bayesian optimization does not merely exploit local optima but actively expands exploration into rewarding regions of the action space. This high-quality initialization proves crucial for subsequent policy learning, ensuring the RL agent trains on meaningful trajectories rather than noise, thereby directly mitigating the spurious reward problem described in our motivation.

*Table 5.* Reward distribution and diversity of generated actions for different sampling strategies.

| Sampling Method | Reward Proportion | Diversity |
|---|---|---|
| | *Exploratory Strategy* | |
| Random Sampling | $0.004 \pm 0.064$ | $2.983 \pm 41.221$ |
| | *Optimized Strategies* | |
| Offline Rollout | $0.093 \pm 0.291$ | $1.608 \pm 13.873$ |
| Our Bayesian | $\mathbf{0.236 \pm 0.493}$ | $1.712 \pm 17.892$ |

**Training Stability and Convergence.** Figure 5 illustrates the impact of reward densification and data augmentation on training dynamics. We track the mean reward scores averaged over rollout trajectories collected via vLLM during the GRPO training phase. We observe that the baseline model (Figure 5b), which relies solely on sparse feedback, struggles to extract meaningful learning signals. The reward curve remains stagnant near zero, indicating a failure to escape the initial exploration phase. This empirical evidence confirms that without densification, the MLLM policy is prone to becoming trapped in local optima or suffering from exploration exhaustion. In contrast, by incorporating dense reward signals, our method (Figure 5a) demonstrates a steady ascent in the early training stages and stabilizes at an asymptotic value of approximately 0.3. The shaded

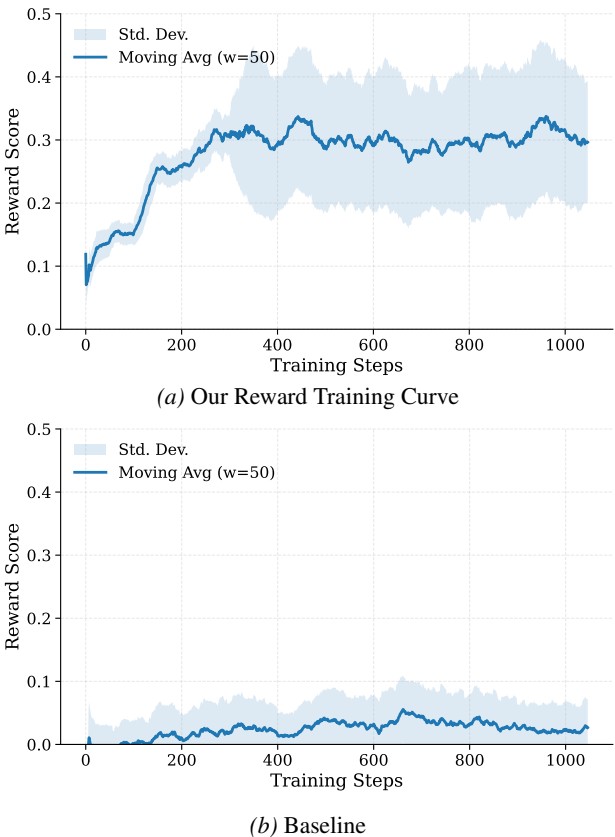

*(a)* Our Reward Training Curve

*(b)* Baseline

*Figure 5.* Impact of Bayesian Reward Modeling on GRPO Training. We visualize the mean reward of rollout trajectories collected.

region indicates manageable variance, suggesting that our reward shaping mechanism effectively guides the policy update, preventing the training collapse often inherent in sparse-reward settings.

## 5. Conclusion

In conclusion, we present B-Spar, a reward-centric RL framework for training perceptually aligned image retouching agents under sparse human feedback. By mitigating reliance on noisy dense proxies, B-Spar directs optimization toward meaningful improvements via prior-guided trajectory sampling, Bayesian reward modeling to densify sparse feedback, and anchor-regularized policy optimization. Extensive experiments on public benchmarks demonstrate that our method significantly outperforms both AIGC and agent-based baselines in perceptual quality, improving the state-of-the-art by approximately $33.5\%$. These results establish superior performance over existing approaches across fidelity metrics highly correlated with human perception (LPIPS, SSIM), quality assessment (HyperIQA), and aesthetics (ArtiMUSE), while maintaining minimal latency. Furthermore, B-Spar effectively addresses the latency bottlenecks typi-

cal of existing multi-agent systems, establishing itself as a highly efficient and perceptually accurate solution for building reliable tool-using editing agents.

## Limitations

This paper focuses on tool-based image enhancement and retouching, rather than on open-ended semantic editing.

## Acknowledgements

We would like to thank the anonymous reviewers for their constructive feedback and helpful suggestions, which significantly improved the quality of this work.

## Impact Statement

This paper presents work whose goal is to advance the field of machine learning. There are many potential societal consequences of our work, none of which we feel must be specifically highlighted here.

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

# A. Implementation Details

## A.1. Domain knowledge learning.

Pre-trained MLLMs lack the domain knowledge needed to understand the underlying image retouching operations and their associated adjustment values. To address this limitation, we designed a series of challenging problems specifically designed to bridge these knowledge gaps. We found that by solving these challenges, machine learning models can become intelligent agents with expert-level domain knowledge, capable of effectively retouching images. **Tool usage.** As a first step, the Multimodal Large Language Model (MLLM) must visually understand the effect of each individual operation on the image, as well as how this effect varies across different adjustment magnitudes. To this end, we randomly sample an operation $O \in \mathcal{L}$ from a predefined library of image retouching operations $\mathcal{L}$, along with a corresponding adjustment value $\delta$. This operation is then applied to the source image $I_{src}$ to generate the edited image $I_{edit}$. The two images, $I_{src}$ and $I_{edit}$, are concatenated and presented to the MLLM, which is tasked with identifying the operation and the adjustment value—that is, given the image pair, the MLLM should predict the operation and its corresponding degree of change. For each image in the AVA dataset, we randomly sample a tool from our retouching toolbox (detailed in Appendix A.3) and randomly generate parameters within valid ranges. This process yields tuples of: $\langle I_{in}, I_{out}, \text{tool\_name}, \text{tool\_parameter} \rangle$ The we input the image pair $\langle I_{in}, I_{out} \rangle$ along with the corresponding tool name and parameters into a vision-language model to generate textual descriptions of visual differences. Here our System Prompt:

---

**Details are provided in the Appendix.**

**User:**
A comparative description based on two images {description} .Using the given function {function} and its parameters {parameters}, deduce the image editing commands provided by the common user (relatively abstract and general) and the professional user (concise, professional, and accurate) (command length should be within 256 characters). The functions and instructions must be aligned; for example, the "curve" function only adjusts exposure and brightness, not color.
Output format: /Common_users_begin /Common_users_end /Experts_begin /Experts_end.

---

From the visual difference descriptions obtained in Stage 2, we reverse-engineer both common user instructions (abstract and general) and expert-level instructions (concise and professional). Here our System Prompt:

---

**Details are provided in the Appendix.**

**User:**
Based on the given function {function} and parameters {parameters}, input image and output image. Compare the input and output images and describe what operation the given function performed on the image. Please consider the following dimensions:
1. Color
2. Tone
3. Contrast
4. Saturation
5. Brightness
6. Color Temperature (YellowBlue): The color of the real light source, indicating the warmth or coolness of the image
7. Color Cast / Tint (MagentaGreen): The degree to which the image leans towards green or magenta
No need to output your thought process; just output directly. For example: The output image has a warmer color temperature than the input image, but the brightness remains largely unchanged.

---

We generate reasoning chains (CoT) based on user instructions. Randomly selecting between expert or common user instructions, we elicit the model to produce natural thinking processes that lead to the specific tool invocation. Here our System Prompt:

---

### Details are provided in the Appendix.

**User:**
The user input command is: {user_command}, which requires calling the image editing function {function} on the input image, with the relevant parameter being {parameter}.
Please deduce your thought process, avoiding any mention of the userprovided function. The thought process and the function being called should be aligned; for example, the "curve" function only adjusts exposure and brightness, not color. The text length should be limited to 256 characters.

---

**Image Aesthetic Understanding.** We used Qwen-2.5-VL-72B as the teacher model to generate detailed inference processes, Qwen-3-VL-30B as the CoT quality judge, and Qwen-2.5-VL-7B as the student model to achieve efficient deployment on the aesthetic dataset ava.

---

### Details are provided in the Appendix.

```
system prompt:
You are an expert in image aesthetic assessment. You will be provided
with an image and a target aesthetic score from a human expert, with
a score range of 0-100. Your task is to analyze the aesthetic
attributes of the image and add a reasoning process to the target
aesthetic score. For each image, first output your reasoning process
and end with the target aesthetic score. Your reasoning should be
detailed, systematic, and well-supported.

user prompt:
Before providing the final score, output your thought process.
Evaluate the image based on fundamental principles of aesthetics
and design. Consider factors like:

Composition (e.g., rule of thirds, leading lines, symmetry, balance)
Color Theory (e.g., complementary, monochromatic, analogous)
Lighting and Shadow
Subject and Focus
Texture and Detail
Originality and Creativity
Emotional Impact
Overall Assessment

Expert's Aesthetic Score: {score}
End your response with 'Final score: {score}'.
```

---

### A.2. Domain knowledge learning setting.

We implemented the Supervised Fine-Tuning (SFT) process using the SWIFT framework (Zhao et al., 2025). The model is initialized with `Qwen3-VL-2B-Instruct` and trained using `bfloat16` precision. To achieve parameter-efficient fine-tuning, we employed Low-Rank Adaptation (LoRA) targeting all linear modules, with a rank of $r = 512$ and a scaling factor $\alpha = 512$. To preserve the pre-trained visual capabilities, both the vision transformer (ViT) and the aligner were frozen during training. The training was conducted on a node with 6 GPUs using DeepSpeed ZeRO-3 optimization to manage memory consumption. The input sequence length was capped at 5,000 tokens, with specific limits set for visual inputs (2,048 tokens for images and 128 tokens for videos with a maximum of 16 frames). We used a global batch size of 96 (calculated as 8 per device × 6 devices × 2 accumulation steps) and an initial learning rate of $1 \times 10^{-4}$ with a 5% warmup ratio. Flash Attention 2 was enabled to accelerate the training process.

### A.3. Tool Definitions

Our retouching toolbox comprises 10 professional-grade image adjustment tools:

---

**Details are provided in the Appendix.**

**1. Curve Adjustment (`call_curve`)** Adjusts image brightness via tone curves. Parameters include `points_8` (8 control points defining the curve) and optional `mask_area` for local adjustments.

**2. Color Adjustment (`call_color`)** Adjusts color temperature and tint. Parameter `color_preset` includes:

- `temperature`: $[-100, 100]$, positive for warmer (yellow), negative for cooler (blue)

- `tint`: $[-100, 100]$, positive for magenta, negative for green

- `vibrance`: $[-100, 100]$, natural saturation adjustment

- `saturation`: $[-100, 100]$, overall saturation

**3. Brightness Adjustment (`call_brightness`)** Controls exposure and tonal range. Parameter `brightness_preset` includes:

- `exposure`: $[-100, 100]$, overall exposure

- `contrast`: $[-100, 100]$

- `highlights`: $[-100, 100]$, bright area adjustment

- `shadows`: $[-100, 100]$, dark area adjustment

- `whites`: $[-100, 100]$, white point adjustment

- `blacks`: $[-100, 100]$, black point adjustment

**4. HSL Adjustment (`call_hsl`)** Hue-Saturation-Lightness adjustment for 8 color ranges (red, yellow, green, cyan, blue, purple, magenta). Supports `standard` and `ipt` modes. Parameter ranges: $[-100, 100]$ for hue, saturation, and lightness.

**5. Calibration (`call_color`)** Camera color calibration correcting color casts. Parameters include:

- `colorspace`: `srgb` or `adobergb`

- `mode`: `xy`, `cone`, `xy_circular`, `xy_circumcenter`, or `mixer`

- `params`: Per-channel (red, green, blue) hue and saturation adjustments

**6. Halation (`call_halation`)** Simulates film halation (light bloom). Parameters: `mode` (`origin`, `standard`, `strong`) and `intensity` controlling blend strength.

**7. Sharpening (`call_sharpen`)** Edge-aware sharpening with skin protection. Parameter: `sharpen` $\in [0, 100]$.

**8. Brilliance (`call_brilliance`)** Enhances shadow details, local contrast, and highlight softness. Parameter: `brilliance` $\in [-100, 100]$.

**9. Vibrance (`call_vibrance`)** Intelligent saturation boosting with skin-tone protection. Parameter: `vibrance` $\in [-100, 100]$.

**10. Color Clarity (`call_color_clarity`)** Enhances chromatic sharpness in LAB color space without affecting luminance. Parameter: `clarity` $\in [0, 100]$.

---

### A.4. Detailed Analysis

Some more specific analyses and numerical results are provided below.

## Details are provided in the Appendix.

**Analysis on the margin/threshold** $\delta$. We conducted a sensitivity analysis by varying $\delta$. We observe that performance remains highly stable:

- $\delta = 1$: $68.75 \pm 4.48$

- $\delta = 2$: $69.02 \pm 4.07$

- $\delta = 3$: $68.69 \pm 4.61$

**On the Hyperparameters** $\sigma_a, \sigma_h$.

| $\sigma_a$ | $\sigma_h$ | **HyperIQA** |
|---|---|---|
| 1.0 | 0.5 | **68.96±4.77** $*$ |
| 0.5 | 1.0 | **68.67±4.35** $*$ |
| 1.0 | 1.0 | **69.02±4.07** $*$ |
| 1.0 | 2.0 | 67.52±5.48 |
| 2.0 | 1.0 | 67.84±8.18 |
| 2.0 | 2.0 | 65.80±7.39 |

(Note: $*$ indicates the stable performance range)

**Reward vs. Action-distance Correlation Analysis.**

| **Dist** | **Reward Rate** |
|---|---|
| 0.05 | 0.977± 0.149 |
| 0.10 | 0.887± 0.316 |
| 0.20 | 0.642± 0.479 |
| 0.50 | 0.123± 0.467 |
| 0.80 | 0.002± 0.007 |

As shown above, reward consistency decreases monotonically with action distance (from 0.977 at distance 0.05 to near 0 at 0.80).

**The use of RGB intensity histograms and more features for image descriptor.** We adopted RGB histograms as a compact descriptor because our current action space focuses on global retouching (e.g., tone and color), where such statistics provide a lightweight yet stable reward signal under limited binary feedback. As shown in the table below, replacing RGB histograms with semantically richer CLIP features yields only marginal improvement, while incurring higher computational overhead. This suggests that for the editing tasks in this study, RGB-based features are surprisingly effective and do not significantly limit the model's performance compared to high-level semantic embeddings.

| **Image Features** | **HyperIQA** |
|---|---|
| $RGB$ | 69.02 ±4.07 |
| $CLIP$ | 69.13 ±4.18 |

**Different kernel functions**

We compared different kernel functions as below:

| **Diffusion Methods** | **HyperIQA** ($\mu \pm \sigma$) |
|---|---|
| $Gaussian$ | 69.02 ±4.07 |
| $Laplace$ | 68.63 ±6.70 |
| $Cauchy$ | 67.00 ±5.13 |

Gaussian achieves the best performance ($69.02 \pm 4.07$), while Laplace and Cauchy show slightly lower performance and/or higher variance. This suggests that while alternative heavy-tailed kernels are applicable, the Gaussian kernel offers a more favorable bias–variance trade-off in our setting.

