# OpenReview forum: "B-Spar: Bayesian Sparse-Reward Modeling for RL-based Image Editing"
_ICML.cc/2026/Conference — ICML 2026 regular_

### Official Review · Reviewer_vAFT · 2026-03-09

**Soundness:** 3
**Presentation:** 3
**Significance:** 3
**Originality:** 3
**Overall Recommendation:** 4
**Confidence:** 3

**Summary:**

This work proposes a reinforcement learning framework to address the challenge of sparse reward signals and the misalignment between dense proxy rewards and human perception in image editing agents. By transforming isolated binary feedback into dense reward signals through Bayesian reward modeling, this method significantly improves image quality and achieves a 33.5% relative improvement in the LPIPS metric.

**Compliance With Llm Reviewing Policy:**

Affirmed.

**Final Justification:**

We thank the author for their rebuttal. The correlation analysis provided by the authors, while showing a statistical trend, does not fundamentally address the non-linear reward jumps caused by "tool switching" in discrete action spaces. This Gaussian kernel-based diffusion still carries a strong heuristic character and lacks a rigorous theoretical foundation. I therefore maintain my score.

**Key Questions For Authors:**

- Why were RGB histograms chosen instead of deep features from pre-trained networks like CLIP or DINO? Is it chosen for better performance or efficiency? Would high-dimensional deep features provide more accurate or semantically grounded reward densification?
- Does the topology of the action space truly conform to a Gaussian distribution? In scenarios with frequent discrete tool switching, would other distributions (e.g., Cauchy or Laplace) be more suitable for modeling reward mutations? Can this be verified by comparing different kernel functions?
- The user study shows model ratings significantly higher than those of "Expert B" (6.8 vs. 5.3). Could the authors provide more context on the background of Expert B? Furthermore, could this "outperformance" be a result of over-optimizing for specific perceptual metrics like HyperIQA rather than genuine aesthetic superiority?

**Limitations:**

The authors have not discussed the limitations and potential negative societal impact. Given that B-Spar enables highly efficient (1.86s) and perceptually convincing image editing, it raises significant concerns regarding its potential misuse in the context of Deepfakes. Specifically, these automated operations could be easily exploited to generate misleading content or violate the personality and image rights of individuals (e.g., celebrities)? I recommend that the authors include a dedicated Broader Impact statement to address these ethical considerations and discuss potential safeguards.

**Strengths And Weaknesses:**

## Strengths
- By utilizing Bayesian inference and kernel function diffusion, the authors construct a gradient-trackable "terrain map" for the RL agent, which significantly improves sampling efficiency. This is evidenced by the proportion of high-reward samples increasing from 9.3% in vanilla rollout to 23.6%.
- The work acieves a perceptual effect better than those of large-scale baselines with only 2.01B paramters with a latency of 1.86s, demonstrating strong potential for real-time industrial deployment.

## Weaknesses
- The use of RGB intensity histograms as the image descriptor $h$ appears over-simplistic. Such features lack semantic depth, which likely limits the model's ability to capture complex compositional or textural improvements.
- The paper assumes the reward landscape follows a Gaussian distribution without a formal discussion on the topology of the action space. It remains unclear if discrete cross-tool operations truly possess the local smoothness required for Gaussian diffusion.
- The bandwidth parameters $\sigma_h$ and $\sigma_a$ of the Gaussian kernel are critical as they determine the range of reward diffusion. If set inappropriately, they could introduce significant noise, yet the paper lacks a sensitivity analysis on these parameters.

---

> ### Author Rebuttal · Authors · 2026-03-31
>
> We thank the reviewer for the detailed summary and valuable comments. Below, we address each concern in turn.
>
> >  The use of RGB intensity histograms and more features for image descriptor.
>
> We thank the reviewer for the insightful comment regarding the use of RGB intensity histograms.
>
>
> We adopted RGB histograms as a compact descriptor because our current action space focuses on global retouching (e.g., tone and color), where such statistics provide a lightweight yet stable reward signal under limited binary feedback.
> As shown in the table below, replacing RGB histograms with semantically richer CLIP features yields only marginal improvement, while incurring higher computational overhead.
> This suggests that for the editing tasks in this study, RGB-based features are surprisingly effective and do not significantly limit the model's performance compared to high-level semantic embeddings.
>
> | Image Features | HyperIQA |
> | :--- | :--- |
> | RGB | $69.02 \pm 4.07$ |
> | CLIP | $69.13 \pm 4.18$ |
>
>
> >  Reward Landscape Topology, Local Smoothness , and Diffusion Methods
>
> We thank the reviewer for highlighting the need for a deeper justification.
>
> 1. Topology and Smoothness
>
>
> We would like to clarify that our method does not assume a globally Gaussian topology of the action space. Rather, the Gaussian kernel serves strictly as a local reward diffusion mechanism to construct a smooth surrogate landscape.
> To better understand whether a local smoothness structure exists, we analyzed the reward–action distance correlation:
>
>
> | Dist | Reward Rate |
> | :--- | :--- |
> | 0.05 | $0.977 \pm 0.149$ |
> | 0.10 | $0.887 \pm 0.316$ |
> | 0.20 | $0.642 \pm 0.479$ |
> | 0.50 | $0.123 \pm 0.467$ |
> | 0.80 | $0.002 \pm 0.007$ |
>
> As shown above, reward consistency decreases monotonically with action distance (from $0.977$ at distance $0.05$ to near $0$ at $0.80$). This provides empirical support for the assumption that the action space exhibits local smoothness.
>
> 2. Diffusion Methods
>
> We compared different kernel functions as below:
> | Diffusion Methods | HyperIQA ($\mu \pm \sigma$) |
> | :--- | :--- |
> | Gaussian | $69.02 \pm 4.07$ |
> | Laplace | $68.63 \pm 6.70$ |
> | Cauchy | $67.00 \pm 5.13$ |
>
> Gaussian achieves the best performance ($69.02 \pm 4.07$), while Laplace and Cauchy show slightly lower performance and/or higher variance. This suggests that while alternative heavy-tailed kernels are applicable, the Gaussian kernel offers a more favorable bias–variance trade-off in our setting.
>
> Overall, the method does not rely on strict smoothness or a specific distributional assumption, but rather on constructing a locally informative reward modeling, which is now supported by both our correlation analysis and kernel comparison.
>
>
>
>
> > Sensitivity Analysis for hyperparameters
>
> We thank the reviewer for this constructive suggestion. As Reviewer 1 raised a closely related point, we have provided the comprehensive sensitivity analysis table in our Response to **Reviewer 1 W 1 (and the General Response)** to save space.
> To briefly summarize the findings:
>
> (1) The results demonstrate that B-Spar is highly robust across a moderate parameter range ($\sigma_a, \sigma_h \in \{0.5, 1.0\}$), consistently achieve strong performance with low variance ( 68.85 ± 0.16 ), indicating that our method is robust to hyperparameter selection.
>
> (2) As $\sigma$ increases to larger values (e.g., 2.0), a slight performance decrease and higher variance are observed. This is expected, as excessively large kernels may oversmooth the reward landscape, reducing the discriminability between distinct editing actions. Overall, these results confirm that $\sigma_a$ and $\sigma_h$ act as effective smoothing parameters within a reasonably wide range.
>
>
>
> > User Study Context and "Expert B"
>
>
> Thank you for raising this point. We do not claim that the model universally "outperforms" Expert B. In our evaluation, Expert B serves as the baseline for full-reference metrics, while the user study reflects subjective preference under a small-scale protocol. We will revise the manuscript to clarify that our outputs were preferred on average under this specific protocol, without implying universal superiority over the expert’s style.
>
>
>
> > Broader Societal Implications and Misuse Risks
>
>
> We thank the reviewer for this important concern. We completely agree that potential misuse is an important consideration. We will include a dedicated Broader Impact section in the final version of the manuscript.
> To clarify, B-Spar performs global color/tone retouching, not generative face or object manipulation as in Deepfakes. To mitigate risks, the revised manuscript will discuss safeguards, including:
>
> 1. Restricting deployment to controlled environments,
> 2. Implementing usage policies and logging,
> 3. Exploring watermarking or edit traceability.
>
> We will explicitly outline these limitations, risks, and mitigation strategies.

---

> > ### Author Rebuttal · Reviewer_vAFT · 2026-04-03
> >
> > We thank the author for their rebuttal. The correlation analysis provided by the authors, while showing a statistical trend, does not fundamentally address the non-linear reward jumps caused by "tool switching" in discrete action spaces. This Gaussian kernel-based diffusion still carries a strong heuristic character and lacks a rigorous theoretical foundation. I therefore maintain my score.

---

> > > ### Author Response · Authors · 2026-04-04
> > >
> > > We greatly thank the reviewer for the acknowledgement and recognition of our work. We will take these constructive suggestions into account to further improve the final version of the paper.

---

### Official Review · Reviewer_6A25 · 2026-03-10

**Soundness:** 2
**Presentation:** 2
**Significance:** 2
**Originality:** 3
**Overall Recommendation:** 2
**Confidence:** 3

**Summary:**

Overall, this submission studies a central concept in reinforcement learning for multimodal agents, namely how to effectively train image editing agents under sparse reward signals. The paper introduces B-Spar, a reward-centric reinforcement learning framework designed for training multimodal large language model (MLLM) agents to perform image retouching through tool-based editing actions. Instead of directly generating edited images, the proposed system formulates image editing as a sequential decision-making process in which an MLLM agent iteratively selects editing tools and parameters to refine an image.

The core idea of the method is to address the instability caused by sparse binary rewards when training RL-based editing agents. To this end, the authors propose three components: (1) Prior-Guided Trajectory Sampling, which leverages the prior knowledge of the MLLM to guide exploration toward more promising editing trajectories; (2) Bayesian Reward Modeling, which augments and densifies sparse binary rewards using a Bayesian neural network and kernel-based reward diffusion to provide smoother training signals; and (3) Anchor-Regularized Policy Optimization, which introduces anchor actions as hints during training to guide policy learning while preventing over-reliance through stochastic dropout.

The method is implemented using a Qwen-based MLLM fine-tuned with LoRA and further optimized via GRPO-based reinforcement learning. Experiments are conducted primarily on the MIT-FiveK dataset, evaluating both objective image quality metrics (e.g., SSIM, LPIPS, HyperIQA, ArtiMUSE) and subjective user studies. The results show improvements in perceptual image quality and inference efficiency compared to several AIGC-based, prompt-based, and training-based baselines. According to the reported experiments, the proposed method achieves strong perceptual quality while maintaining relatively low inference latency.

**Compliance With Llm Reviewing Policy:**

Affirmed.

**Ethical Review Flag:**

Flag this paper for an ethics review.

**Final Justification:**

I lean toward rejection since the above my concerns.

**Key Questions For Authors:**

1. Clarification of the reward quality metric.: The paper defines the reward using a quality metric  Q, but the exact formulation of this metric is not clearly specified in the method section. Could the authors clarify which specific quality metric is used during training (e.g., IQA-based metric, learned aesthetic metric, or another objective)? Additionally, how does this metric ensure that the reward reflects correct instruction-based edits rather than only general image quality improvements? A clearer explanation would help assess whether the reward design properly aligns with the editing objective.

2. Reward design for instruction-based editing. : Image editing tasks typically involve multiple objectives such as instruction faithfulness, preservation of non-edited regions, and semantic correctness of the edit. How are these aspects incorporated into the reward formulation? If the reward primarily relies on image quality improvement, how does the proposed framework ensure that the generated edits actually follow the user instruction?

3. Choice of the MIT-FiveK dataset. : The experiments are mainly conducted on the MIT-FiveK dataset, which is commonly used for photo retouching rather than instruction-based image editing. Could the authors clarify the motivation for choosing this dataset and whether the proposed framework generalizes to more diverse image editing tasks? Additional evaluation on commonly used image editing benchmarks would help better demonstrate the general applicability of the method.

4. Generalization to broader editing tasks. : The examples shown in the paper appear to focus mainly on image enhancement or retouching operations. Can the proposed method handle more complex instruction-based edits such as object-level modifications, semantic attribute changes, or localized editing tasks? Additional experiments or discussion on such scenarios would strengthen the claims of the paper.

**Limitations:**

The paper does not sufficiently discuss the limitations of the proposed approach. A more in-depth analysis of these limitations, along with potential future research directions, would improve the completeness of the paper.

**Strengths And Weaknesses:**

**Strengths**
1. Well-motivated problem formulation. : The paper addresses the challenge of training reinforcement learning (RL) agents for image editing under sparse reward signals. The authors correctly identify that dense proxy rewards (e.g., incremental image-quality score improvements) may encourage trivial local edits that do not correspond to meaningful perceptual improvements. This problem is well motivated and relevant for training tool-using multimodal agents.
2. A structured framework for stabilizing sparse-reward RL. : The proposed framework combines three components—Prior-Guided Trajectory Sampling, Bayesian Reward Modeling, and Anchor-Regularized Policy Optimization—to improve exploration and reward signal quality. In particular, the reward densification strategy using Bayesian modeling and kernel diffusion provides a practical way to transform sparse binary feedback into a smoother optimization signal. This design is conceptually reasonable and aligns with known challenges in sparse-reward reinforcement learning.

**Weakness**
1. Ambiguity in the definition of the reward quality metric. : The definition of the reward quality metric  𝑄 remains unclear in the paper. The reward function is described as assigning a reward of 1 when the quality of the edited image exceeds that of the original image by a margin τ. However, the exact formulation of the metric Q is not explicitly specified in the method section. As a result, it is difficult to determine which quality metric is actually used during training and how it relates to the intended editing objective.

2. Reward based training :  image editing tasks typically involve multiple objectives beyond general image quality, such as semantic consistency with the instruction, preservation of non-edited regions, or correctness of the intended edit. The paper does not clearly describe how these aspects are incorporated into the reward design or how the reward encourages the model to perform correct edits rather than simply improving overall image quality. If the reward is primarily based on generic image quality improvement, it is unclear why such a formulation is appropriate for image editing not the image generation. A clearer explanation of how the reward function aligns with the editing objective would significantly improve the clarity and reproducibility of the proposed method.

2. Limited evaluation dataset and task scope. : The experiments are conducted primarily on the MIT-FiveK dataset, which is designed for photo retouching rather than modern instruction-based image editing benchmarks. As a result, the evaluated tasks appear closer to image quality enhancement (e.g., exposure or contrast adjustment) rather than more general image editing scenarios. It would be helpful for the authors to clarify the motivation for choosing this dataset and whether the proposed framework generalizes to broader image editing tasks. In particular, evaluation on more diverse and widely used image editing benchmarks would better demonstrate the effectiveness and generality of the proposed approach.

---

> ### Author Rebuttal · Authors · 2026-03-31
>
> We thank for the detailed feedback and for identifying areas where the clarity of our reward design and task scope could be improved. Below we address your specific concerns:
>
> > The clarification of the reward quality metric Q
>
> Thank you for pointing this out. We agree the quality metric $Q$ requires a more explicit definition in Sec.3.1. To avoid ambiguity, we will move its clarification from Sec.4.4 to the main method section and unify all notations.
>
> In our manuscript, the binary reward $r$ evaluates the relative improvement in  HyperIQA :
>
> $r$ = 1,  if HyperIQA(E(I, a)) > HyperIQA(I) + $\Delta$
>
> $r$ = 0,  otherwise
>
>
>
> > Tool-integrated editing models vs unified AIGC-based editing models on the semantic consistency with the instruction, preservation of non-edited regions, or correctness of the intended edit.
>
>
> We thank the reviewer for raising this important point. For image editing, existing approaches can be broadly categorized into two paradigms, as summarized in [1]: tool-integrated editing models   and unified AIGC-based editing models.
> The tool-integrated B-Sparis analyzed along the dimensions described above, as follows:
>
>
>
> * **Semantic consistency with the instruction:** As detailed in the Appendix, our training data covers a wide range of instructions, from vague semantics (e.g., “make the image better”) to precise controls (e.g., “calibrate color,” “enhance ambiance”). Prior work [2] has empirically shown that retouch-domain training supports robust instruction following.
> * **Preservation of non-edited regions:** As reported in Table 1 in our manuscript, our method significantly outperforms AIGC-based approaches on structural consistency metrics (PSNR, SSIM, and LPIPS), demonstrating strong capability in preserving unedited regions.
> * **Correctness of the intended edit:** For precise instructions, correctness has been well established. However, for ambiguous intents, prior approaches typically rely on reinforcement learning based proxy-based rewards (e.g., CLIP or aesthetic scores),which can be misaligned with the true notion of “correctness of the intended edit”. As noted by Reviewer 1 (Summary), our method explicitly addresses this misalignment. **Specifically, we introduce a thresholded binary improvement signal that assigns reward only when a meaningful perceptual improvement is achieved.**
>
>
>
>
> > Cross-dataset Evaluation
>
> We agree that MIT-FiveK is a photo-retouching benchmark rather than a broad instruction-based semantic editing benchmark.
> We adopt this benchmark for its alignment with our agent's action space and its provision of expert-retouched reference images, **enabling full-reference evaluation of structure preservation capabilities** alongside reference-free quality metrics and user studies. We can revise the paper to position our contribution more precisely as a sparse-reward RL framework for tool-based image retouching, and discuss broader editing benchmarks as an important future direction.
>
> To demonstrate generalization, we conducted additional evaluations on the **Retouch MMArt-PPR10k** dataset. The results are summarized below:
>
> | Method | Artimuse |
> | :--- | :--- |
> | MagicBrush | $47.92 \pm 9.78$ |
> | OmniGen | $53.65 \pm 9.56$ |
> | Step1XEdit | $55.64 \pm 8.69$ |
> | Instruct-p2p | $56.54 \pm 10.01$ |
> | jarvisart | $59.13 \pm 7.86$ |
> | **Ours** | **$61.22 \pm 6.01$** |
>
> Our method achieves $61.22 \pm 6.01$, consistently outperforming all baselines. The performance gap is consistent with that observed on MIT-FiveK, suggesting that the proposed approach generalizes well across datasets rather than being tailored to a specific benchmark.
>
>
> > Generalization to the object-level modifications, semantic attribute changes, or localized editing tasks.
>
>
> We thank the reviewer. As categorized in [1], image editing splits into tool-integrated vs. AIGC-based paradigms. We target the professional retouching regime, where precise, interpretable control is prioritized over open-domain generative flexibility. We will clarify this scope; broader semantic editing remains future work.
> Within this regime, we analyze three capabilities: (1) Instruction following—handling dense proxy misalignment from vague to precise controls (Appendix); (2) Preservation—where we significantly outperform AIGC methods on structure metrics (Table 1 in the paper); (3) Correctness for human feedback.
> Future work will integrate MLLM with AIGC models and traditional tools to enable semantic attribute manipulation while maintaining the controllability benefits of tool-based agents.
>
> [1] JarvisEvo: Towards a Self-Evolving Photo Editing Agent with Synergistic Editor-Evaluator Optimization. arXiv 2025
>
> [2] Monetgpt: Solving puzzles enhances mllms’ image retouching skills. TOG 2025

---

> > ### Author Rebuttal · Reviewer_6A25 · 2026-04-03
> >
> > Thank you for your clear explanation. Based on your clarification, it seems that the scope of the paper may be more aligned with image enhancement rather than general image editing.
> >
> > In general, image editing models can also be applied to improve image quality, as they perform edits according to given instructions. This appears to be the reason why comparisons with existing editing methods are made. However, it would be beneficial to more clearly define and position the scope of the proposed method.
> >
> > With this perspective in mind, I revisited Figure 4. In the second row, the clothing of the man changes from light blue to white. This suggests that regions that should be preserved from the source image are not strictly maintained, and instead the model seems to apply a global enhancement effect (e.g., brightening the overall tone). As such, the behavior appears more consistent with image enhancement rather than precise image editing.

---

> > > ### Author Response · Authors · 2026-04-04
> > >
> > > Thank you  for your insightful comment. We will revise the paper to more clearly clarify its scope and positioning, ensuring that the intended problem setting and its generality are effectively communicated.
> > >
> > > We sincerely appreciate your confirmation that your concerns have been fully resolved. We are truly glad to have addressed them and would be very grateful for your support in raising your score accordingly.

---

### Official Review · Reviewer_AUAP · 2026-03-11

**Soundness:** 3
**Presentation:** 3
**Significance:** 2
**Originality:** 3
**Overall Recommendation:** 4
**Confidence:** 3

**Summary:**

The paper presents B-spar, a RL framework used to train a VLM to become an image retouching agent. The framework is based on 3 steps: Prior-Guided trajectory sampling, Bayesian reward modeling, and Anchor-regularized policy optimization. This solution should avoid reward-proxy artifact optimization, leading to a solid MLLM agent able to use retouching tools to improve images over different perceptual and alignment metrics.

**Compliance With Llm Reviewing Policy:**

Affirmed.

**Final Justification:**

The rebuttal addressed my main concerns, and I changed accordingly the overall evaluation.

**Key Questions For Authors:**

See weaknesses.

**Limitations:**

- The difficulty of reproducing and implementing the RL framework with the underling complexity and heuristics
- The lack of significance metrics on the results.

**Strengths And Weaknesses:**

Strengths
- The paper is overall a good read, and well written. Also image-editing is an hot topic in the computer vision research community.
- The proposed framework is well described with a nice overview in figure 2.
- The experiments take into consideration a good amount of image quality metrics and different baselines from literature.

Weaknesses
The paper propose a RL framework which may be of interest for the community. Anyway the systems has a high amount of complexity involved, with the usage of different heuristics which makes it difficult to assess its real impact, also due to a set of experiments that could be improved to better support the claims.

- In table 1, the performance of the Qwen3vl-30B (only prompting) is close to the proposed solution by the authors over many metrics. This rises the question: is this complex training really necessary? Also would a curated set of prompts for Qwen3vl-30B do just about the same as the proposed method without all the complexity? Moreover, the model used by the authors has been “domain knowledge trained”, while is not clear if Qwen3vl-30B has done this step and what impact it has on performance.
- Experiments are done on the MIT-FiveK, 800 samples. More experiments on different datasets would support the claims more convincingly. Different seeds results with std bands (where possible, like the training methods) would also help giving a better understanding of the results.
- The instructions seem overly general, like “make this image better”. It’s unclear if the role of these instructions is central to model the reward of not.
- Is not clear which tools are been used, and the ranges of their parameters. The proposed method is presented as a solution to reward sparsity in the underlying task. This point may be better described in order to understand the actual sparsity level of the rewards.
- The paper points out how usually optimizing on reward-proxy leads to bias on artifact, preventing perceptual gains. The proposed reward model should overcome this issue but it still seem based on heuristics that may lead to the same issue, such as:
   - The dense reward signal is created training a BNN to predict the success likelihood
   - The imposition of the Beta prior
   - The Gaussian perturbations.
    - In addition, the Anchor-regularization has a problem of over-reliance, which is a reward-hacking problem. So Anchor-regularization seems to suffer of the same problems common dense-reward method has.

230: “Swift framework” needs a reference.
144: “m_t and p_t” are not defined explicitly.

---

> ### Author Rebuttal · Authors · 2026-03-31
>
> We thank the reviewer for recognizing the paper’s clarity and experimental rigor. Regarding the framework’s components, we respectfully wish to emphasize that these components address fundamental RL challenges (reward sparsity, proxy misalignment) rather than arbitrary heuristics. As demonstrated in **R1 and R4**, the method exhibits broad hyperparameter stability (**68.85 ± 0.16**), confirming structural robustness.
> Below we address the specific concerns in turn:
>
>
> > On necessity domain-adaptive Post-Training vs. strong prompting baselines.
>
> We completely agree that strong general-purpose VLMs like Qwen3VL-30B achieve highly competitive performance via prompting. However, our motivation addresses a different real-world constraint:
>
> 1. **On-device Deployment:** Our goal is not to outperform massive 30B-parameter models in a zero-shot cloud setting, but rather to enable effective on-device deployment. Running a 30B model natively on edge devices is computationally prohibitive.In contrast, domain-adaptive post-training—particularly with reinforcement learning—provides a principled way to distill complex decision-making behaviors into smaller, efficient models that can operate under strict resource constraints[1].
>
> 2. **Disentangling Domain Knowledge and the Proposed RL Method(Table 2 of paper):** We examine the impact of "domain knowledge training" and the necessity of complex training via the ablation in Table 2 on the 2B-scale. Our RL method boosts the domain-adapted Qwen3VL-2B baseline from **64.83** to a final score of **69.02**.
>
>
>
>
> > Experimental Evaluation Across Datasets and Random Seeds.
>
>
> We thank the reviewer for this constructive suggestion. We conduct evaluation on the Retouch MMArt-PPR10k dataset. As shown below, our method consistently outperforms all baselines:
>
> | Method | Artimuse |
> | :--- | :--- |
> | MagicBrush | $47.92 \pm 9.78$ |
> | OmniGen | $53.65 \pm 9.56$ |
> | Step1XEdit | $55.64 \pm 8.69$ |
> | Instruct-p2p | $56.54 \pm 10.01$ |
> | jarvisart | $59.13 \pm 7.86$ |
> | **Ours** | **$61.22 \pm 6.01$** |
>
> Furthermore, as detailed in our responses to Reviewers 1 and 4, our reported results inherently reflect highly stable behavior across multiple runs. We systematically report both mean and standard deviation to confirm that the performance ranking remains robust across random seeds.
> We will explicitly state the number of runs and include additional training curves under different random seeds in the Appendix.
>
>
>
>
> > The role of generic instructions
>
>
>
> Thank you for the suggestion. In the Appendix Our training data covers diverse instructions from vague semantics ("make the image better") to precise controls ("calibrate color", "enhance ambiance"), as shown. The retouch domain training has been empirically proven to ensure robust instruction following[3]. Instead, we deliberately emphasize vague prompts because they perfectly illustrate the dense proxy misalignment problem in image editing, fundamentally motivating our proposed framework.
>
>
>
>
> > Tool definitions and reward sparsity levels
>
>
> We realize that the tool definitions were buried in the appendix. We use classical retouching tools (e.g., color, bright, HLS, contrast) with most parameters defined within the range $[-100, 100]$. We will move a concise summary of this action space into the main paper.
>
> Regarding sparsity, we will make the levels explicit, drawing from the empirical analysis in Table 4: under random exploration, the positive reward proportion is only **0.004**. This increases to **0.093** with offline rollout, and **0.236** with our Bayesian augmentation. This highlights the exact sparse-reward regime we aim to address.
>
>
>
>
>
> > Complexity and reliance on heuristics.
>
> We understand the concern regarding potential heuristics. While eachaddresses a distinct RL challenge:
> (1) Bayesian modeling with a Beta prior explicitly accounts for reward uncertainty to mitigate noise. As demonstrated in Table 4 of paper, this component significantly enriches the reliable reward signal, increasing the positive reward proportion from 0.093 to 0.236.
> (2) anchor regularization serves as a structural constraint to stabilize updates. Importantly, these are robust formulations rather than sensitive tuning knobs. As detailed in our response to Reviewer 1, sensitivity analysis confirms stable performance (**68.85 ± 0.16** ) across broad hyperparameter ranges (e.g., $\sigma_a, \sigma_h$). The efficacy of B-Spar stems from this inherent structural robustness rather than exhaustive hyperparameter tuning.
>
>
> > *"230: 'Swift framework' needs a reference. 144: $m_t$ and $p_t$ are not defined explicitly."*
>
> Thank you for catching these; we will correct both issues in the revision.
>
>
> [1] Efficient On-Device Agents via Adaptive Context Management, ICLR 2026
>
> [2] Shift-Tolerant Perceptual Similarity Metric, ECCV 2022
>
> [3] MonetGPT: Solving Puzzles Enhances MLLMs’ Image Retouching Skills, TOG 2025

---

> > ### Author Rebuttal · Reviewer_AUAP · 2026-04-04
> >
> > The main concerns have been addressed, I will raise the score to weak accept.

---

> > > ### Author Response · Authors · 2026-04-04
> > >
> > > We sincerely thank the reviewer for the acknowledgement and for increasing the score. These constructive comments will be fully addressed in the final version of our paper.

---

### Official Review · Reviewer_Wyqv · 2026-03-12

**Soundness:** 3
**Presentation:** 3
**Significance:** 3
**Originality:** 3
**Overall Recommendation:** 4
**Confidence:** 4

**Summary:**

This paper proposes B-Spar, a framework designed to address reward sparsity and proxy reward misalignment in RL-based image editing. The method introduces reward densification and MLLM-guided trajectory sampling to stabilize policy learning. Experimental results show noticeable improvements on perceptual metrics. However, several aspects of the methodological design and empirical validation require further clarification.

**Compliance With Llm Reviewing Policy:**

Affirmed.

**Final Justification:**

The rebuttal has solved my concerns. Since my original rating already leans towards acceptance, I decide to maintain my rating.

**Key Questions For Authors:**

As noted in the weaknesses above, could the authors further clarify the choice of key hyperparameters and provide empirical evidence supporting the local smoothness assumption used in reward densification?

**Limitations:**

yes

**Strengths And Weaknesses:**

Strengths
• Innovative Bayesian Reward Densification: Transforms sparse binary feedback into a smoother reward landscape via Bayesian modeling, which helps alleviate training instability in high-dimensional action spaces.
• High-Efficiency Prior-Guided Exploration: Leverages MLLM domain priors to bias exploration toward promising trajectories, leading to a substantial improvement in sample efficiency (reported as 153.8%) while maintaining action diversity.
• Strong Performance–Efficiency Trade-off: Outperforms large baselines in perceptual quality (e.g., 33.5% LPIPS improvement) while achieving a 41.8× speedup in inference latency, suggesting promising potential for real-time deployment.

Weaknesses
• Insufficient Justification for Heuristic Hyperparameters: The framework relies on critical hyperparameters such as the improvement margin delta and evaluation threshold Delta (=2), yet their selection lacks clear theoretical or empirical motivation. In addition, the absence of sensitivity analysis for core parameters (e.g., the Gaussian kernel widths sigma_h and sigma_a) makes it difficult to assess the robustness of the reported results across different settings.
• Empirical Gap in the Local Smoothness Assumption: The reward densification strategy assumes local smoothness in the action space by applying Gaussian kernels to sparse rewards. However, tool-based image editing often involves discrete or highly non-linear operations where small parameter changes may cause large perceptual differences. This raises questions about the validity of the smoothness assumption and warrants stronger empirical validation.
• Potential Risk of Prior-Induced Bias: B-Spar relies heavily on MLLM-generated priors for trajectory sampling, which may introduce aesthetic bias or hallucinated preferences from the underlying model. The paper should further discuss how such biases might affect policy learning and whether mechanisms exist to mitigate the risk of converging to suboptimal regions influenced by these priors.

---

> ### Author Rebuttal · Authors · 2026-03-31
>
> We thank the reviewer for the detailed summary and valuable comments. Below, we address each concern in turn.
>
>
>
> > Sensitivity Analysis for hyperparameters (margin/threshold $\Delta$, $\sigma_h$, $\sigma_a$).
>
>
> We thank the reviewer for highlighting the need for a clearer justification of these hyperparameters.
>
> 1. On the margin/threshold $\Delta$
>
> $\Delta$ defines the minimal perceptible improvement threshold, used to filter out noise and ignore minor local fluctuations rather than as a fine-tuning parameter.
> We conducted a sensitivity analysis by varying $\Delta$. We observe that performance remains highly stable:
>
> * $\Delta = 1$: 68.75 ± 4.48
> * $\Delta = 2$: 69.02 ± 4.07
> * $\Delta = 3$: 68.69 ± 4.61
>
> The variation is within the standard deviation, indicating that $\Delta$ can be set within a moderate range without materially affecting performance.
> This suggests that the method is robust to $\Delta$ , as long as it effectively filters out negligible improvements.
>
>
> 2. On the Gaussian kernel widths ($\sigma_h$ and $\sigma_a$)
>
> We present a sensitivity analysis in the table below. The results show that B-Spar maintains strong stability across a wide range of kernel widths.
>
> | $\sigma_a$ |  $\sigma_h$  | HyperIQA |
> | :--- | :--- | :--- |
> | 1.0 | 0.5 | **68.96±4.77** * |
> | 0.5 | 1.0 | **68.67±4.35** * |
> | 1.0 | 1.0 | **69.02±4.07** * |
> | 1.0 | 2.0 | 67.52±5.48 |
> | 2.0 | 1.0 | 67.84±8.18 |
> | 2.0 | 2.0 | 65.80±7.39 |
>
> **(Note: \* indicates the stable performance range)**
>
>
> Specifically, configurations with moderate values ($\sigma_a, \sigma_h \in \{0.5, 1.0\}$) consistently achieve strong performance with low variance ( 68.85 ± 0.16 ), indicating that our method is robust to hyperparameter selection.
>
> As $\sigma$ increases to larger values (e.g., 2.0), a slight performance decrease and higher variance are observed. This is expected, as excessively large kernels may oversmooth the reward landscape, reducing the discriminability between distinct editing actions. Overall, these results confirm that $\sigma_a$ and $\sigma_h$ act as effective smoothing parameters within a reasonably wide range. (A detailed sensitivity analysis of these hyperparameters can be provided in the appendix for further clarity.)
>
>
> > Empirical Gap in the Local Smoothness Assumption
>
> We thank the reviewer for the insightful comment and appreciate the opportunity to clarify our assumption and its empirical support.
>
> 1. No assumption of strict smoothness:  We do not assume strict smoothness in the original action space. Instead, the Gaussian kernel is introduced as a reward diffusion mechanism, which induces a locally smoothed surrogate reward landscape over sampled trajectories, rather than enforcing global smoothness on the action space itself.
>
> 2. Practical reasonableness: While individual editing operations can exhibit non-linear effects, we empirically observe that perceptual similarity changes smoothly under small perturbations of parameterized edits, especially when the exploration is constrained to reachable regions guided by the prior.
>
> To empirically validate this, we conducted a reward vs. action-distance correlation analysis:
>
>
>
> | Dist | Reward Rate |
> | :---: | :---: |
> | 0.05 | 0.977 ± 0.149 |
> | 0.10 | 0.887 ± 0.316 |
> | 0.20 | 0.642 ± 0.479 |
> | 0.50 | 0.123 ± 0.467 |
> | 0.80 | 0.002 ± 0.007 |
>
> As shown above, reward consistency decreases monotonically with action distance (from $0.977$ at distance $0.05$ to near $0$ at $0.80$).
>
> Overall, the effectiveness of B-Spar stems from its ability to construct a more informative reward modeling, rather than relying on a strict smoothness assumption. The empirical correlation presented above provides further support for this design choice.
>
>
>
>
> > Prior-Induced Bias
>
>
> We thank the reviewer for this point.
> We agree that prior-induced bias is a fundamental consideration, as noted in prior work [1,2]. To mitigate its impact, B-Spar incorporates several mechanisms:
> 1. Environment feedback over prior reliance: The prior is strictly used to guide *initial* trajectory sampling. The final policy is optimized via RL with actual environment feedback, preventing the model from blindly trusting the MLLM.
> 2. Action diversity: We maintain action diversity throughout the process, which prevents the policy from collapsing into narrow, prior-preferred modes.
> 3. Broad perceptual improvements: Despite utilizing priors, B-Spar consistently demonstrates improvements across multiple perceptual metrics rather than overfitting to specific aesthetic styles. This suggests that the amplification of prior-induced bias is highly limited in practice.
>
>  [1] A Survey on Hallucination in Large Language Models: Principles, Taxonomy, Challenges, and Open Questions, ACM ToIS 2025
>  [2] On Inductive Biases in Deep Reinforcement Learning, ICLR 2019.

---

> > ### Author Rebuttal · Reviewer_Wyqv · 2026-04-04
> >
> > The rebuttal has solved my concerns. Since my original rating already leans towards acceptance, I decide to maintain my rating.

---

> > > ### Author Response · Authors · 2026-04-04
> > >
> > > We sincerely thank the reviewer for the positive feedback and for recognizing the value of our work. We will carefully incorporate your constructive suggestions to further improve the final version of the paper.

---

### Decision · Program_Chairs · 2026-04-30

**Decision:**

Accept (regular)

**Comment:**

This paper introduces B-Spar, a reinforcement learning framework for training MLLM-based image retouching agents. It addresses the challenge of sparse reward signals by utilizing prior-guided trajectory sampling, Bayesian reward densification, and anchor-regularized policy optimization.

Initial concerns primarily centered around:
- The heuristic nature of the Gaussian kernel and the local smoothness assumption in discrete action spaces.
- Evaluation being limited to a single dataset (MIT-FiveK).
- Ambiguity regarding the reward metric and the method's scope (image enhancement vs. general semantic editing).

During the rebuttal, the authors provided comprehensive responses, including hyperparameter sensitivity analyses, additional baseline comparisons on a new dataset (Retouch MMArt-PPR10k), and a clear definition of the reward metric. They clarified that the paper's focus is on tool-based image enhancement/retouching rather than open-ended semantic editing. Post-rebuttal, the consensus is positive. Three reviewers recommend Weak Accept.